# Real-Time Phaseless Microwave Frequency-Diverse Imaging with Deep Prior Generative Neural Network

**Zhenhua Wu** [1,2,3], **Fafa Zhao** [1], **Man Zhang** [4,*], **Jun Qian** [1] and **Lixia Yang** [1]

1   Information Materials and Intelligent Sensing Laboratory of Anhui Province, Anhui University, Hefei 230601, China
2   East China Research Institute of Electronic Engineering, Hefei 230031, China
3   State Key Laboratory of Millimeter Waves, Southeast University, Nanjing 210096, China
4   School of Electronics and Communication Engineering, Guangzhou University, Guangzhou 510006, China
*   Correspondence: mzhang401@gzhu.edu.cn

**Abstract:** The millimeter-wave frequency-diverse imaging regime has recently received considerable attention in both the security screening and synthetic aperture radar imaging literature. Considering that the minor systematic errors and alignment errors could still produce heavily corrupted images, these complex-based imaging reconstructions rely heavily on the precise measurement of both phase and amplitude of radiation field patterns and echo signals. In the literature, it is shown that by leveraging phase-retrieval techniques, salient reconstruction images can still be acquired, even in the presence of significant phase errors, which could ease the phase error calibration pressure to a large extent in practical imaging applications. In this paper, in the regime of phaseless frequency-diverse imaging, with the powerful feature inference and generation power of unsupervised generative models, an end-to-end deep prior generative neural network is designed to achieve near real-time imaging. The harsh imaging reconstruction with both the high radiation mode correlations and extremely low scene compression sampling ratio, which are extremely troublesome to tackle for generally applied matched-filter and compressed sensing approach in the current frequency-diverse imaging literature, can still be preferably handled with our reconstruction network. The well-trained reconstruction network is constituted by prior inference and deep generative modules with excellent generative capabilities and significant prior inference abilities. Using simulation experiments with radiation field data, we verify that the integration of phase-free frequency-change imaging with deep learning networks can effectively improve reconstruction capabilities and improve robustness to systematic phase errors. Compared with existing imaging methods, our imaging method has high imaging performance and can even reconstruct targets under low compression ratio conditions, which is somewhat competitive with current state-of-the-art algorithms. Moreover, we find that the proposed method has good anti-noise and stability.

**Keywords:** real-time imaging; frequency-diverse imaging; phaseless imaging; deep prior generative network

## 1. Introduction

Frequency-diverse imaging has gained popularity in both metasurface antenna design and synthetic aperture radar imaging [1–3] applications in recent trends. In principle, the capacity to shape complicated radiative wave-fronts and couple energy from the reference wave to the desired radiation pattern offers the potential to generate diverse beam patterns with less hardware complexity than traditional electronic beam steering arrays. The key feature of frequency-diverse imaging is that the scene information is encoded onto a collection of spatially distinct radiation field patterns and reconstructed under a forward imaging model. Despite that data acquisition in frequency-diverse imaging is performed in an all-electronic manner, and no mechanical scanning is required, the complex-based scene image reconstructions still rely on the precise measurement of both phase

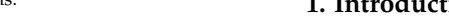

and amplitude of the echo signal, as any tiny antenna positional misalignments and field pattern characterizations would degrade the phase coherence accuracy of an imaging system, yielding phase error calibration techniques [4,5] to be leveraged.

In general, systematic errors including misalignment of the transmitter and receiver apertures predominantly introduce phase error into the system, and antenna modulation errors can contribute further to phase errors. In [6], various misalignment types, including displacement, rotation, and direction on the performance of a sparse, bi-static, frequency-diverse imaging system, are elaborately studied, and the impact of phase errors are quantitatively evaluated. In [7], phase-retrieval techniques are leveraged to minimize the impacts of phase calibration and alignment errors on image reconstruction. Given that the complex-value-based image reconstruction places heavy requirements on accurate system characterization of the RF cables and connectors, the technique to relax the phase coherency requirement would be the integration of the frequency-diverse technique with phase-retrieval techniques, as demonstrated in [8]. Note that phase retrieval has been shown to be a viable technique in conventional microwave and millimeter-wave SAR imaging and diagnostics applications [9,10].

In more recent research trends, a large body of literature on programmable computational meta-images used as trainable physical layers for data interpretation have been demonstrated [11–14]. One such approach, proposed in [12,13], is based on the radiation of illumination patterns specific to different types of scenes to be imaged, necessarily requiring the use of reconfigurable systems for scene-dependent beam synthesis. Provided prior knowledge about the nature of the scene to be imaged is available, these works have shown that it is possible to limit the number of sequential captures necessary for image reconstruction compared with the use of purely random patterns. Recent work in [14] took this idea even further by directly integrating a model of the physical layer into an artificial neural network in order to jointly learn optimal measurement and processing strategies based on a priori knowledge of scene, task, and measurement constraints. Since this "learned sensing" strategy enables one to minimize the acquisition of task-irrelevant information, it is highly task-specific and requires a supervised learning technique.

Note that the primary challenge of the generally applied compressed sensing (CS) [15] reconstruction techniques in the current frequency-diverse imaging literature is that the success of the reconstruction process depends heavily on the randomness as well as the condition number of the measurement matrix. Furthermore, when it comes to the imaging scenario of extremely large dimensionality, the computation costs will drastically increase given the needed memory and storage usage for matrix inversion and iteration. CS techniques with an ill-conditioned measurement matrix still suffer from the prohibitively high signal-to-noise ratio to maintain a preferable imaging performance.

Given that the deep learning technique has been broadly used in various applications, such as image super-resolution, phase recovery, and scattering medium imaging [16–22], a cascaded complex U-net (CCU-Net) model was put up in [23] as a solution to the complex domain of phaseless-data inverse scattering problems (PD-ISPs). In [24], the method that combines concepts from the projected gradient descent approach for solving linear inverse problems using generative priors and the alternating minimization (AltMin) approach for nonconvex sparse phase retrieval was proposed, and an analysis of sample complexity using Gaussian measurements was provided to verify the performance. In [25], a generative prior-assisted compressive phase-retrieval algorithm was proposed, and the effectiveness for physically realizable coded diffraction pattern measurements in low measurements and high noise regimes was demonstrated.

In this paper, we marry the concept of deep prior generative neural networks with the phaseless frequency-diverse imaging technique. By employing a prior feature extractor network [26], the underlying structural regularity of the target image is effectively learned. Furthermore, with the knowledge of the extracted prior features, the generator network [27] can produce images with high-quality texture structures. In particular, the scene image amplitude data are described by the probability distribution defined by the generative

network model; considering the observation data still contain the information of the original target, the latent features describing the basic laws of the original image are extracted from the compressed target amplitude data. To further manifest these extracted features, a latent variable with a Gaussian distribution is employed, and the output of the generator network is accessed with multiple convolutional layers.

With the goal of quickly converging to the local minima, the Adaptive Momentum Estimation (ADAM) optimizer [28,29], with a tailored learning rate and relatively low memory usage, is used to update the network weight parameters. Both the complicated target FMNIST [30] data and the widely known sparse target MNIST [29] dataset are utilized to train the network. The trained reconstruction network is therefore able to directly recover the scene reflectivities with high fidelity and handle the large dimensionality of the measurement matrix. The effectiveness and reliability of our PFDI-Net method is confirmed through extensive imaging simulations using the collected radiation field data.

Traditional frequency-diverse computational imaging methods, such the CS and SBL methods, suffer from issues with sluggish target reconstruction speed and inadequate use of measurement data. Aside from that, the effect of phase error and phase calibration brought on by improper recovery of acquired phase data on the initial target scene image with echo data amplitude is also taken into consideration. In our present work, phaseless imaging is leveraged to relax the phase coherency requirements of imaging system; deep prior generative neural networks are designed to perform scene image reconstruction, and the capability of the inference network could effectively extract the prior information from the collected echo data and assist the generative network to resolve the scene image. In this model, the information in the observations, which is disregarded in earlier models, influences the learning of the characteristics and the creation of the original images.

Additionally, the suggested model can recover the test data through quick mapping rather than iterative optimization thanks to inferred networks in order to completely utilize the remaining information in the observations, to produce a more well-structured output, and to restore the original image for the reconstructed imaging problem of huge scene targets. In contrast to the existing works [23–25,31], the approach we propose here applies deep networks to the sensing matrix rather than the specific expected scene. Consequently, our approach is not scene-dependent and does not require the use of sequential measurements relying on active reconfigurable antennas. We propose a system-dependent but scene-independent method relying on a frequency sweep to generate a succession of random illumination patterns that interrogate the scene to be imaged. Since the antenna measurement modes are frequency-dependent, the superior image reconstruction capability could therefore reduce the need for large operation bandwidth and, more importantly, the antenna radiation efficiency could be enhanced in the antenna design process to some extent so as to maintain a relatively high SNR level, which is crucial for near-field computational imaging. Via the optimized deep neural reconstruction network, the dimensionality of the measurement matrix can be limited, and thereby the computational complexity of the image reconstruction can be reduced; in addition, from the aspect of imaging system implementation, the large needed operation frequency band and optimal designing burden of current metasurface antennas' front-end could be eased to some extent.

## 2. Imaging Principle

The frequency-diverse imaging system based on metasurface antennas initially analyzes the entire working mechanism of the system before constructing a suitable mathematical model. The basic concept of frequency-diverse imaging is to construct a subwavelength resonant aperture structure to regulate the polarization characteristics of electric and magnetic dipoles and produce different polarization characteristics by designing subwavelength basic resonant units with different geometric structures. As the driving frequency changes, the radiation field excited from different subsets of resonators changes. Objects within the scene scatter the incident fields, producing the backscattered components detected by the waveguide probe at the transmitting antenna plane.

The frequency measurements collected through the receiving probe are related to the scene reflectivities by the measurement matrix (transfer function) constituted by the product of the electric fields from the transmitting antenna and the receiving probe at each position in the scene. The schematic diagram of a metasurface antenna imaging system with a single transmitter and receiver is shown in Figure 1. The following proportionality applies to all fields that propagate into the OEWG:

$$g(f) \propto \int_V \mathbf{E_{TX}}(\vec{\mathbf{r}}'; f) \cdot \mathbf{E_{RX}}(\vec{\mathbf{r}}'; f) \cdot \sigma(\vec{\mathbf{r}}') \, \mathrm{d}^3\vec{\mathbf{r}}' \tag{1}$$

where the transmitted and received fields are denoted as $\mathbf{E_{TX}}(\vec{\mathbf{r}}')$ and $\mathbf{E_{RX}}(\vec{\mathbf{r}}')$, respectively. Scene target reflectivities are represented by $\sigma$. The conventional way of solving for an estimate of the scene reflecectivity vector, $\sigma_{est}$, involves using various computational imaging algorithms, from the single-shot matched filter technique to iterative compressed-sensing-based methods. Different from complex-based, leveraging phase-retrieval techniques, $\sigma_{est}$ is reconstructed from intensity-only measurements of the signal scattered from the object, $|g|^2$. Considering the imaging system is both diffraction- and bandwidth-limited, a more compact measurement equation to reconstruct $\sigma$ can be expressed as:

$$|g|^2 = |H\sigma + n|^2 \tag{2}$$

The intensity-only measurements of the scene could significantly reduce the effect of phase errors on image reconstruction. In Equation (2), $g \in \mathbb{R}^{M \times 1}$ is the intensity-only measurements of the signal scattered from the object, and $n \in \mathbb{R}^{M \times 1}$ represents the noise vector. In the simplest fashion, assuming additive white Gaussian noise, a matched filter (MF) reconstruction suitably solves (2) as $|\hat{\sigma}|^2 = |H^\dagger g|^2$, where † denotes the conjugate transpose operator. Additionally, more advanced reconstruction algorithms such as the least squares algorithm and regularization methods are also widely used. In general, these direct and iterative image reconstruction techniques rely heavily on the matrix inversion, in which the measurement mode orthogonality and scene sampling ratio play a key role. Additionally, a minor perturbance induced by phase error could drastically damage the image reconstruction. In this consideration, in the framework of phaseless imaging, image reconstruction is performed with a deep reconstruction neural network; the effect of measurement mode correction and extremely low scene sampling ratio on image reconstruction can be reduced in comparison with conventional complex-based, matrix-inversion-based reconstruction techniques.

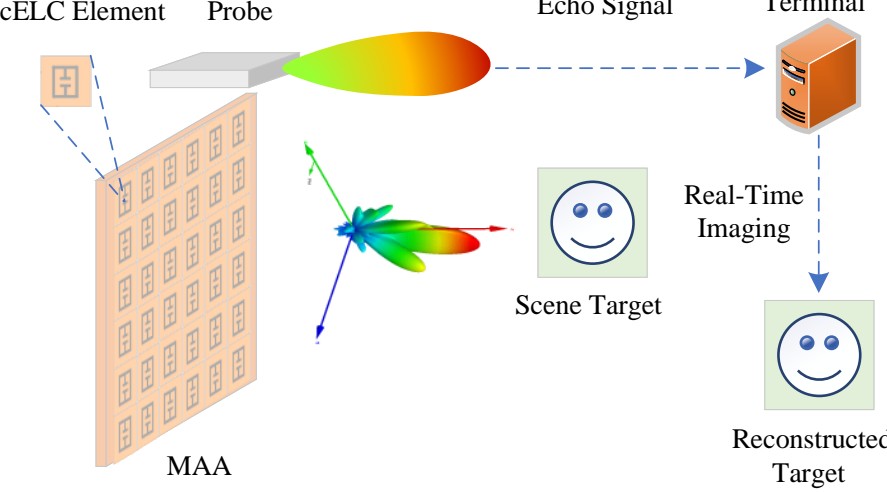

**Figure 1.** Schematic diagram of the frequency-diverse imaging system.

## 3. Imaging Network Model

In the deep learning literature, GAN has recently performed well in describing the prior distribution of imaging targets. The network is capable of generating very clear and realistic images, and it is even difficult for the human eye to judge whether the images are real or fake. At the same time, VAE is also a very effective generative model, which can extract latent features of descriptive data through an effective inference network. This paper integrates these two models well, which makes the model proposed in this paper have its own advantages. The characteristics of the two are introduced separately below.

### 3.1. Feature Inference Network

As a form of deep generative models, a variable autoencoder is a generative network structure based on variable Bayesian inference, proposed by Kingma et al. in 2014. Unlike traditional autoencoders that describe the bit space numerically, it describes the observations of the bit space in a probabilistic way, showing great application value in data generation.

A variational autoencoder essentially learns the hidden relationship between the input variable $x$ and the hidden variable $z$. Given $x$, the conditional probability distribution of the hidden variable is $p(x|z)$. After learning this distribution, different samples can be generated by sampling $p(x|z)$.

From a probabilistic point of view, we assume that any dataset is sampled from some distribution $p(x|z)$. Z is a hidden variable representing some internal feature. For example, pictures $x$ and $z$ of a handwriting dataset can represent some settings, such as bootstrap size, writing style, bold, italics, etc., which fit some prior distribution $p(z)$. Given a particular hidden variable $z$, we can sample a series of generated samples from the learned distribution $p(x|z)$ that have a commonality denoted by $z$.

When the distribution $p(z)$ is known, we want to learn to generate a probabilistic model $p(x|z)$; here, we can use maximum likelihood estimation: a good model should have a good chance of producing the observed samples $x \in \mathbb{D}$. If the generative model $p(x|z)$ is parameterized with $\theta$, for example, we learn $p(x|z)$ through a neural network that is a decoder, then $\theta$ is the weights $w$, $b$, etc., of this decoder, so the optimization goal of the neural network is:

$$\max_{\theta} p(x) = \int_z p(x|z)p(z)dz \tag{3}$$

Since $z$ is a continuous variable, the above integral cannot be converted to a discrete form, making it difficult to directly optimize the above formula. Using the idea of variational inference, use the distribution $q_{\phi}(z|x)$ to approximate $p(x|z)$, that is, the distance between the two needs to be optimized:

$$\min_{\phi} \mathbb{D}_{KL}(q_{\phi}(z|x)|p(x|z)) \tag{4}$$

KL divergence $\mathbb{D}_{KL}$ is a measure of the gap between distributions $q$ and $p$, defined as:

$$\mathbb{D}_{KL}(q|p) = \int_x q(x) \log \frac{q(x)}{p(x)} dx \tag{5}$$

Strictly speaking, the distance is generally symmetric, while the KL divergence is not, and the KL divergence is expanded as:

$$\mathbb{D}_{KL}(q_{\phi}(z|x)|p(z|x)) = \int_z q_{\phi}(z|x) \log \frac{q_{\phi}(z|x)}{p(z|x)} dz \tag{6}$$

For the optimization objective function $\mathcal{L}(\theta, \phi)$, our goal is to maximize the likelihood probability $\max p(x)$ or $\max \log p(x)$, so we can use $\max l(\theta)$ to achieve this. As shown in the following formula:

$$\mathcal{L}(\theta, \phi) = \int_z q_\phi(z|x) \log \frac{p_\theta(x,z)}{q_\phi(z|x)} = -\mathbb{D}_{KL}(q_\phi(z|x)|p(z)) + \mathbb{E}_{z \sim q}[\log p_\theta(x|z)] \qquad (7)$$

Therefore, the encoder network parameterizes the $q_\phi(z|x)$ function, and the decoder network parameterizes the $p(x|z)$ function, it can be obtained by computing the KL divergence between the decoder output distribution $q_\phi(z|x)$) and the prior distribution $p(z)$ and by the loss function of the likelihood probability $\log p(x|z)$ between the decoder targets to optimize $\mathcal{L}(\theta, \phi)$.

### 3.2. Deep Generative Adversarial Network

With the development of deep learning, people hope to find richer hierarchical models to better describe the probability distribution of various complex data encountered in real life. By far the most successful applications of deep learning are models that map high-dimensional informative sensor inputs to class labels. These remarkable successes are mainly based on backpropagation and dropout algorithms, which use piecewise linear functions and can generate particularly efficient gradients. Because deep generative models will suffer from some problems, such as many complex high-dimensional probability calculations, it is difficult to approximate in maximum likelihood estimation and similar learning strategies, and it is even more difficult to use piecewise linear functions in the generation process to generate efficient gradient.

In the proposed generative adversarial network framework, the generative model competes in the training phase with an adversary, which is a discriminative model, which determines whether a sample comes from the generative model's distribution or the learned data distribution. The generative model can be viewed as a counterfeit-like team trying to produce counterfeit money and deceive the police by using it; the discriminative model is like the police, trying to detect counterfeit money. In the process of confrontation, both sides try to improve their methods by learning until they can not tell the difference between the real and the fake.

For the generator to learn to describe the distribution $p_g$ of the data $x$, define a prior distribution $p_z(z)$ of the input noise variable, and then denote a mapping from $z$ to the data space as $G(z; \theta_g)$. Moreover, define another multilayer perceptron $D(x; \theta_d)$ capable of outputting a single scalar. $D(x)$ represents the probability that $x$ comes from the data rather than the generator $G(\cdot)$ by training $D(\cdot)$ to maximize the probability of assigning the correct label to the generated samples from $G(\cdot)$ and samples from the real data. At the same time, $G(\cdot)$ is trained to minimize $\log(1 - D(G(z))$, that is, by simultaneously training $D(\cdot)$ and $G(\cdot)$ to play the following two-player max-min game $V(G, D)$:

$$\min_G \max_D V(G, D) = E_{X \sim P_{data}(x)}[\log D(x)] + E_{z \sim p_z(x)}[\log(1 - D(G(z)))] \qquad (8)$$

Essentially, under nonparametric conditions, the training criterion enables the generative distribution $G(\cdot)$ to accurately describe the data distribution when $D(\cdot)$ is sufficiently discriminative. In practical applications, iterative numerical methods are generally used to implement the game, and the optimization process usually takes $l$ steps $D(\cdot)$ optimization and one step $G(\cdot)$ optimization. Alternating between the two results in that as long as $G(\cdot)$ changes slowly enough, $D(\cdot)$ can stay near its optimal solution.

### 3.3. PFDI-Net: Architecture and Training

For VAE, when it is simply used to generate images, the generated images are more regular but blurred; for GAN, its training process is not so stable, and it is prone to problems such as mode collapse or gradient disappearance. In order to solve the respective problems

of VAE and GAN, this paper adopts the combined use of the two models to give full play to the advantages of the two models to compensate for their respective shortcomings.

From the perspective of VAE, the target generated by VAE is relatively vague, and a large part of the reason is that it does not know how to better define the loss between the generated target and the real target. The traditional VAE will define the loss by comparing the pixel difference between the generated target and the real target, and then take the mean value, which results in the generated target being blurred. To solve this problem, we can add a discriminator to the VAE. At this time, when the decoder of the VAE generates the target, not only should the loss between the generated target and the original target be small but also the generated target should be deceived by the discrimination. The addition of the discriminator forces the decoder of the VAE to generate clear targets.

From a GAN perspective, the generator of a traditional GAN receives guidance from the discriminator when generating targets, thereby gradually generating realistic targets as training progresses. In a simple GAN structure, however, since the capabilities of the generator and discriminator are difficult to balance, it is easy to cause instability in training. One of the important reasons is that the generator of GAN has never seen a real target and tries to generate a target directly from a large amount of data. At this time, the ability of the generator has difficultly competing with the discriminator to achieve confrontation. In this case, we usually need to adjust the parameters of the generator multiple times or train for a long time to make the model converge. The function of adding a VAE encoder to GAN is to add a loss to the generator, that is, the loss between the generated target and the real target, which is equivalent to telling the generator what the real target looks like, and the generator has an additional loss as a guide; it will be more stable when training.

In the proposed PFDI-Net, as shown in Figure 2, the imaging network mainly consists of three main components: encoder, generator, and discriminator. Specifically, the encoder consists of four fully connected layers with 500, 1000, 500, and 40 neurons, each layer activated by the Relu function. The generator has four convolutional layers with channels of 256, 128, 64, and 1, a corresponding convolution kernel of $3 \times 3$, $4 \times 4$, $4 \times 4$, $4 \times 4$, and a fully connected layer of 784 neurons sent to the convolutional layer after receiving the feature hidden vector extracted by the encoder for feature learning. Four convolutional layers and a fully connected layer form the discriminator. The structure of the generator and discriminator is symmetrical, with the four convolution layers having channels of 1, 64, 128, and 256, and the convolution kernel sizes are all $4 \times 4$. The only fully connected layer helps the discriminator's output discrimination probability. It should be emphasized that Leaky Relu is used in all imaging network layers to activate the function. The Leaky Relu function is a variant of the traditional and well-known Relu activation function; solving the problem of Relu function in the negative interval improves the model training and test fitting ability because the derivative is always nonzero and reduces the development of silent neurons in order to produce a better generation effect. The above content was updated and added in the revised version of the manuscript.

The PFDI-Net proposed by us, and the overall structure of the imaging network model, is shown in Figure 2, which is made up of three primary components—an encoder, a generator, and a discriminator. Where $\sigma$ represents the original scene target, $\hat{\sigma}$ represents the reconstructed scene target, $|g|^2$ represents the echo measurement, $|\hat{g}|^2$ represents the reconstructed echo measurement, and z represents the original scene compressed by the encoder. The hidden vector of the target feature, $\varepsilon$ , represents the discriminator output probability: output 1 is true; output 0 is false. Additionally, the associated loss varies for each component since the generative model is represented by the mapping $G : \mathbb{R}^{K \times 1} \rightarrow \mathbb{R}^{N \times 1}$, where $K$ is the dimension of the feature latent vector and $N$ is the number of grid cells in the imaging space; moreover, $K \ll N$. We assume that the scene target in Equation (3) is $\sigma \in \mathbb{R}^{N \times 1}$ and that the inferred model mapping relationship is $E : \mathbb{R}^{N \times 1} \rightarrow \mathbb{R}^{K \times 1}$. To better describe the latent features of the scene target image data, we introduce a Gaussian random latent variable $z \in \mathbb{R}^{K \times 1}$, so that there will be a generative model: $p_\theta(\sigma, z) = p_\theta(\sigma|z)p_0(z)$. Among it, $p_0(z) \sim N(0, I)$ is the prior distribution of $z$,

which is used to describe the cognition of the data. $p_\theta(\sigma|z)$ is described by a generative network $G(\cdot)$, and $\theta$ contains all the parameters for generating the network.

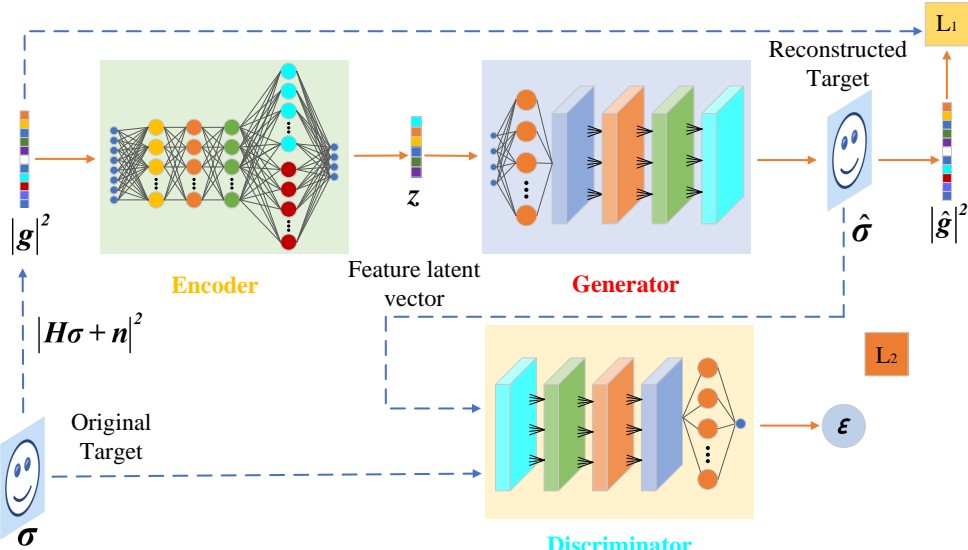

**Figure 2.** Model diagram of PFDI-Net.

With the target scene picture $\sigma \in \mathbb{R}^{N\times1}$ as a starting point, the encoder condenses the inferred features to create a low-dimensional latent vector $z = E(\sigma) \in \mathbb{R}^{K\times1}$, which is then supplied to the generator for imaging training to create a new sample $G(z)$ that closely resembles the target scene. Following the aforementioned training procedure, the model is stored for quick recall of test imaging. We suggest minimizing the following objective function in order to be able to extract information exclusively from the magnitude measurements in (9) and to more accurately reconstruct the target image:

$$\hat{\sigma} = \arg\min\left\| |g|^2 - |H\sigma|^2 \right\|^2 \tag{9}$$

Finding the target $\sigma$ in (10), which ideally only comprises samples taken from the image distribution, with $\sigma = G(z)$, and falling within the range of the generator, is our goal. In the low-dimensional latent representation space, the reduction technique in (10) can be expressed equivalently as follows:

$$\hat{z} = \underset{z\in\mathbb{R}^k}{\arg\min}\left\| |g|^2 - |HG(z)|^2 \right\| \tag{10}$$

The idea behind this optimizer is to modify the latent representation vector z until the generator generates an image $\sigma$ that is consistent with (10). Due to the modular operator and nonlinear deep generative model, the optimization process in (10) is nonconvex and nonlinear. To locate the local minimum $|g|^2$, we use a gradient descent approach. It is important to note that when entering the method as a pretrained model, the generator's weights are always fixed. Through the forward pass of the generator $G(\cdot)$, z is solved to provide the estimated image. The desired result is $\sigma = G(z)$, and the ideal $z$, denoted $\hat{z}$, is the one that has the minimum reconstruction error.

For parametric inference, the true posterior distribution of the latent variable $p(z|\sigma, g) = p_\theta(\sigma, g|z)p_0(z)/(\int p(\sigma, g, z)dz)$ is very complicated. Therefore, it is difficult to obtain the desired explicit solution by expected maximum (EM), mean field variance Bayesian, Monte Carlo sampling (MCMC), and other methods. Furthermore, the number of samples to be processed in compressed sensing tasks is usually very large, which presents a test for the algorithm to handle large sample data. In view of this, the algorithm combines the characteristics of VAE, and effectively solves the parameters of the model by combining the prior inference network $E(\cdot)$ and the generative network $G(\cdot)$.

Inference $q_\phi(z|g)$ through the prior feature network, i.e., the variational distribution of the output of the deep neural network has sufficient statistics to approximate the complex and true posterior distribution $p(z|\sigma)$ of the hidden variable $z$:

$$q_\phi(z|g) = \mathcal{N}(\mu_z(g), diag(\sigma_z^2(g))) \tag{11}$$

where it represents a Gaussian distribution whose mean is $q_\phi(z|g) = \mathcal{N}(\mu_z, diag(\sigma_z^2))$, the diagonal covariance matrix is $diag(\sigma_z^2)$, and the vector $diag(\sigma_z^2)$ is its diagonal elements. Any nonlinear function can be implemented $\mu_z(g)$ and $diag(\sigma_z^2)(g)$, such as a deep neural network, and $\phi$ contains all the parameters of the inference network.

The parameters of the prior inference network $E(\cdot)$ and the generative network $G(\cdot)$ are jointly optimized by the following cost function:

$$\mathcal{L}_1(E, G) = \min_{E,G} \||g| - |HG(E(g))|\|_2^2 + \lambda KL(q_\phi(z|g)||p_0(z)) \tag{12}$$

In Equation (12) above, the first term describes the error between the generated measurements and the actual measurements, while the second term is the prior regularization constraint on the implicit vector $z$, and the KL divergence is used for regularization.

To better enable the generated network to map the hidden variable $z$ to the space of the original image, an adversarial learning method is adopted in this paper. The generator $G(\cdot)$ and the discriminator $D(\cdot)$ train the imaging network alternately with the following min–max cost function:

$$\mathcal{L}_2(E, G) = \min_G \max_D E_{\sigma \sim P_{data}(\sigma)}[\log D(\sigma)] + E_{z \sim q_\phi(z|g)}[\log(1 - D(G(z)))] \tag{13}$$

By alternately optimizing the cost functions $\mathcal{L}_1$ and $\mathcal{L}_2$, the above imaging network model can be well trained by Algorithm 1.

---

**Algorithm 1** PFDI-Net training algorithm.

---

**Input:** $H$: measurement matrix; T: maximum training epochs; $\{g_1, g_2, \cdots g_m\}$ : Echo signal testing dataset; $\{\sigma_1, \sigma_2, \cdots \sigma_n\}$: Scene target training dataset.
1: Initialize $E(\cdot)$, $G(\cdot)$, $D(\cdot)$;
2: Iteration via a gradient descent scheme:
3: **for** T **do**
4:     Sampling a batch of s training samples $\{\sigma_1, \sigma_2, \cdots \sigma_s\}$
5:     For the i-th training sample, calculate $|g_i|^2 = |H\sigma_i|^2$
6:     Regarding the cost function (12), the inference network $E(\cdot)$ and the generative network $G(\cdot)$ are updated by the ADAM optimization algorithm.
7:     The discriminator network $D(\cdot)$ is updated by the ADAM optimization algorithm with respect to the cost function: $-\frac{1}{s}\sum_{i=1}^{s}[\log D(\sigma_i) + \log(1 - D(G(E(g_i))))]$.
8:     The generator network $G(\cdot)$ is updated by the ADAM optimization algorithm with respect to the cost function: $-\frac{1}{s}\sum_{i=1}^{s}[\log(1 - D(G(E(g_i))))]$;
9: **end for**
10: $\hat{\sigma} = G(E(\hat{g}_i))$
**Output:** The target reflection coefficient estimate $\hat{\sigma}$.

---

### 3.4. Measured Field Data

The measured radiation field data in this section serve to validate the proposed PFDI-Net approach. In this portion, a two-dimensional parallel-plate waveguide metasurface antenna with a waveguide slot feeding mechanism is designed and constructed to show the effectiveness of CI. By using the near-field scanning method, or measurement matrix, to measure the radiation field of various frequencies, a distance of 0.5 m must separate the scanning plane from the antenna platform. The image can then be recreated using the measured measurement matrix. Use an open-waveguide (OEWG) probe as the receiving antenna, including the panel-top probe configuration, to provide appropriate backscattered

signal collection from all feasible directions and all frequencies. The antenna panel is 250 mm$^2$ in size, has a dielectric constant of 3.66, and a loss tangent of 0.003. The upper conductor of the waveguide uses 125 × 125 cELC metamaterial resonators, each of which has a Q value between 50 and 60. The substrate thickness between the copper ground layer and the conductive copper metamaterial hole is 0.5 mm. Table 1 displays the system specifications of antennas.

**Table 1.** Main system parameters of metasurface antenna.

| Parameters | Values |
| --- | --- |
| Operation bandwidth | 33∼37 GHz |
| Antenna panel size | 250 × 250 mm$^2$ |
| Number of resonance units | 125 × 125 |
| Frequency sampling interval | 10 MHz |
| Field of view (Azimuth) | −60°∼60° |
| Field of view (Elevation) | −60°∼60° |
| Azimuth sampling interval | 2° |
| Elevation sampling interval | 2° |
| Dimensions of **T** | 400 × 3721 |

The imaging experiment is based on the simulated metasurface antenna radiation field pattern data and imaging scene. The operation bandwidth of the antenna is 33∼37 GHz, the frequency sampling interval is 10 MHz, and the pattern of each frequency point is sampled along the two-dimensional spherical coordinate system of elevation and azimuth. The size of the field of view (FOV) is the elevation angle (−60°∼60°), and the sampling interval is 2°; the azimuth angle sampling line of sight is (−60°∼60°), and the sampling interval is 2°, so the size of the original pattern T is 400 × (61 × 61).

The original target that contains the sparse target and extended target is employed to qualitatively evaluate the imaging ability of the measurement matrix for the scene. In order to qualitatively evaluate the ability of the measurement matrix to image the scene, the image containing the point-scattering target in the same dimension as the measurement matrix is used as the original image. It should be emphasized that the measurement matrix at this time is the pattern data, not the metamaterial in the actual imaging space. The measurement matrix is formed by the radiated field of the aperture antenna.

Figure 3 shows the singular value curves corresponding to the selected measurement matrix in different measurement modes, wherein the measurement modes are 400, 200, and 100 corresponding to M/N of 0.1, 0.05, and 0.025, respectively. In general, the number of nonzero singular values determines the accuracy of scene reconstruction using the pseudo-inverse operation, in the presence of measurement noise. The smaller singular value of the denominator term will diverge when the inversion operation is performed, which makes the reconstructed solution of the matrix inversion seriously deviate from the optimal solution.

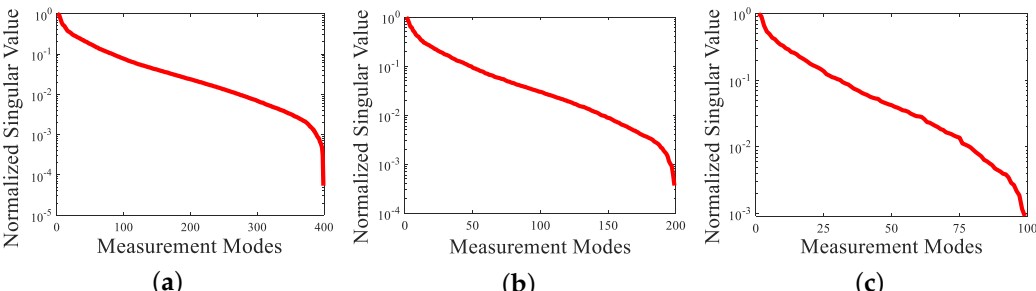

**Figure 3.** Singular values corresponding to the number of different measurement modes in the measurement matrix. (**a**) 400 (M/N = 0.1). (**b**) 200 (M/N = 0.05). (**c**) 100 (M/N = 0.025).

### 3.5. Data Preparation

This paper uses the MNIST and Fashion-MNIST datasets to construct the target datasets required for our simulation tests. The original target scattering coefficients and associated echo measurement magnitudes should be included in the target datasets. Both datasets consist of 60,000 training images and 10,000 testing images, each of size $28 \times 28$. The initial images of the two datasets built above were modified to $61 \times 61$ in accordance with the experimental demands and the actual imaging specifications of the metasurface antenna used in the preceding section. Consider this adjustment to be an image made up of a number of scatter points with random scatter coefficient values between 0 and 1, and think of it as such. In order to create the datasets needed for our imaging, referred to as PFDI-MNIST and PFDI-FMNIST, respectively, 20,000 target images from each of the two datasets mentioned above were chosen. Each dataset contains the amplitude value of the echo measurement and the original target picture. The PFDI-MNIST dataset and the PFDI-FMNIST dataset under imaging settings were generated by the imaging algorithm and were separated into 70% training set, 20% validation set, and 10% test set.

Additionally, the Adam optimization algorithm is employed with a learning rate of 0.0001 and a batch size of 256 to optimize the complete imaging network model utilizing mean square error (MSE) and KL divergence as loss functions, and we employ the imaging quality metrics Peak Signal-to-Noise Ratio (PSNR) and Structure Similarity Index Measure (SSIM), which are noted alongside imaging outcomes. The imaging model is configured to train for 1000 epochs, and the latent vector $z$ dimension is set at 40. It takes a lot of time to validate findings and generate them after each training period, but it is necessary to track network performance. Following the completion of the network training, the network model is saved, the measured scene echo measurement value's amplitude value is fed into the network, and the network model is called to produce the target image in real time.

The batch processing of the above data sets is carried out on the MATLAB platform. The network model is implemented on python 3.7 using tensorflow version 2.5 and is trained on a desktop computer with Nvidia 3070 Ti GPU and CUDA version 11.1. The desktop computer is $\times 64$ compatible, with a Windows 10 64 bit operating system, Intel (R) core (TM) i7-10700 cpu@ 2.90 GHz, and 32 GB memory.

### 3.6. Numerical Tests

The PFDI-Net algorithm that we proposed can adopt a combination of an a priori inference model and generative adversarial model and performs imaging simulation training on the common sparse target image dataset PFDI-MNIST and the extended target dataset PFDI-FMNIST, respectively, to test the performance of our system's target power generation capability. In order to conduct numerical experiments and performance evaluation more logically, we separately selected three targets for imaging comparisons in the above datasets. The measurement mode of the measurement matrix is selected as 400, which is equivalent to a scene information sampling rate M/N of 0.1, and the imaging result is shown in Figure 4.

It should be emphasized that the imaging simulation process, with an emphasis on the application of our suggested approach, takes noise-free scenarios into account throughout. The experimental results demonstrate that the proposed algorithm can extract the prior information of the target contained only by inputting the amplitude value of the echo measurement value and cannot only reconstruct the regular sparse target but can also successfully restore the extension target under the condition of a fixed number of measurement modes and scene compression ratios. Additionally, the algorithm that we proposed can generate high-quality rebuilt images.

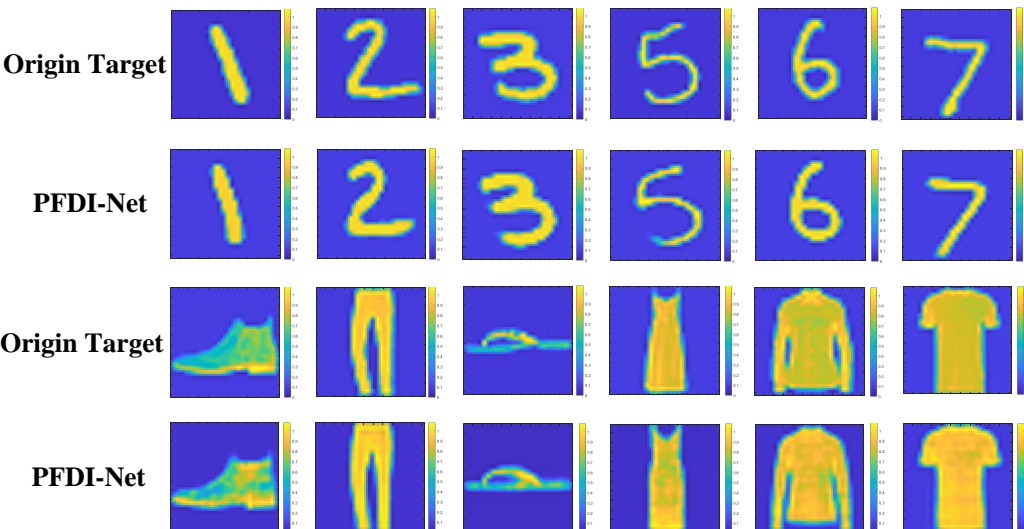

**Figure 4.** Reconstruction results from PFDI-Net with different scene target.

To further assess the efficiency of our suggested approach in reconstructing pictures using just amplitude information in the phaseless state, we ran a number of simulation experiments. First, the performance of our suggested approach is evaluated using the PFDI-Net in sparse and extended target datasets at various scene information compression ratios. The results are displayed in Figures 5 and 6. It can be observed that our model can still provide a reconstruction target with a distinct target contour and good resolution when M/Ns are as low as 0.025, 0.05, and 0.1. Our approach passed the test even when the information compression ratio is lower, demonstrating strong resilience and efficiency.

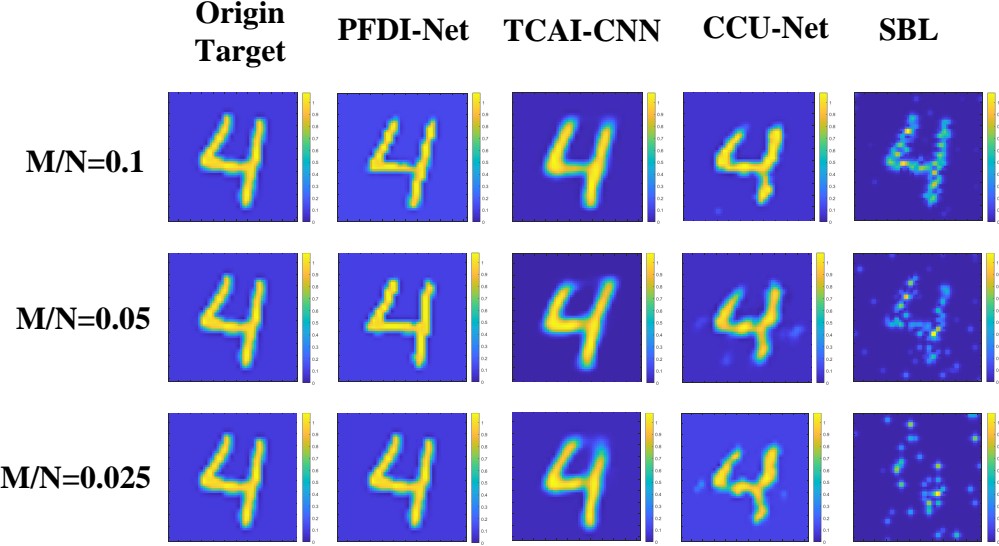

**Figure 5.** PFDI-MNIST: Reconstruction results in four imaging algorithm with different scene information sampling ratios.

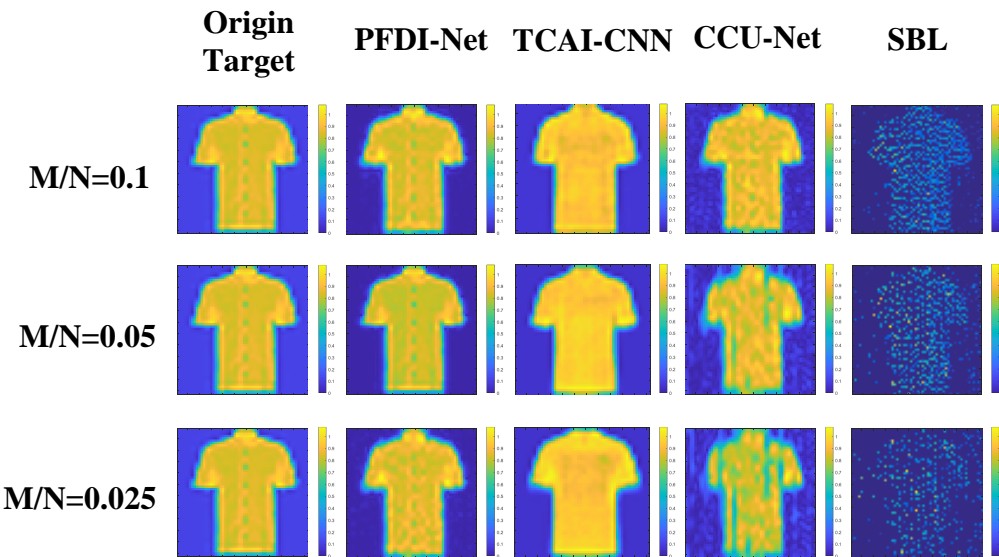

**Figure 6.** PFDI-FMNIST: Reconstruction results in four imaging algorithm with different scene information sampling ratio.

Then, the imaging outcomes of our suggested approach for resolving the inverse problem are contrasted with those of a traditional sparse Bayesian learning (SBL) algorithm, a cascaded complex U-net (CCU-Net) model [23], and a Terahertz Coded-Aperture Imaging network (TCAI-CNN) [21]. In Figures 5 and 6, although the TCAI-CNN reconstruction imaging results show the approximate shape, they cannot capture the detailed features specific to the target. Additionally, the original target contour features cannot be precisely reconstructed in the CCU-Net imaging results, the SBL algorithm is unable to operate in the case of extremely low compression ratios, and the imaging outcomes essentially show no details of the original target. In contrast, the target contour information could be perfectly retrieved with our method, and our approach comes closer to the actual scenario in the target building. In general, in comparison with the other three methods, while the other three approaches can recreate the target's overall shape, the details are still missing and come with some strong artificial points. Our PFDI-Net approach yields higher resolution results with more accurate scattering intensity and a sharper target profile under all M/N situations. Moreover, the imaging quality steadily improves as M/N increases. The algorithm that is proposed by us offers higher resolution results with more realistic scattering intensities and sharper target contours than the other three algorithms under all M/N conditions.

Additionally, the proposed algorithm offers imaging more quickly than the classical SBL algorithm. According to an average of 10 experiments for each approach, Table 2 displays the time needed for various techniques. Given how easily neural-network-based methods may be parallelized, we also monitor the reconstruction time while using GPUs to implement the suggested method. The end-to-end network, on the other hand, can directly translate the echo signal's amplitude value into the target after compression, whereas classical imaging methods need numerous iterations to predict a viable solution. Therefore, it is not unexpected that generative model-based approaches are faster.

**Table 2.** Imaging runtime.

| Methods | Values |
|---|---|
| SBL | 1.02 s |
| CCU-Net [23] | 0.40 s |
| TCAI-CNN [21] | 0.29 s |
| PFDI-Net (CPU) | 0.23 s |
| PFDI-Net (GPU) | 0.04 s |

In order to reflect the imaging performance of the respective algorithms, we further use image Peak Signal-to-Noise Ratio (PSNR) and the Structure Similarity Index Measure (SSIM) value with the reference target image as two criteria to quantitatively evaluate the quality of the recovered targets, as illustrated in Figures 7 and 8; as the M/N varies for PSNR in Figures 7a and 8a, our approach produces greater reconstruction quality and is numerically superior to the other three methods. In terms of SSIM in Figures 7b and 8b, our method is likewise significantly superior, thus attesting to the success of the PFDI-Net approach in reconstructing the intricate scene targets. Compared with previous imaging methods, our PFDI-Net is added to the feature prior inference network model. Due to the learning ability of the prior inference network, it can better learn the real distribution of the targe, and gradually generate realistic targets with training. No matter the M/N, the proposed approach performs better than the SBL, TCAI-CNN, and CCU-Net algorithms. The results demonstrate how effective the PFDI-Net approach is at reconstructing complicated scene items.

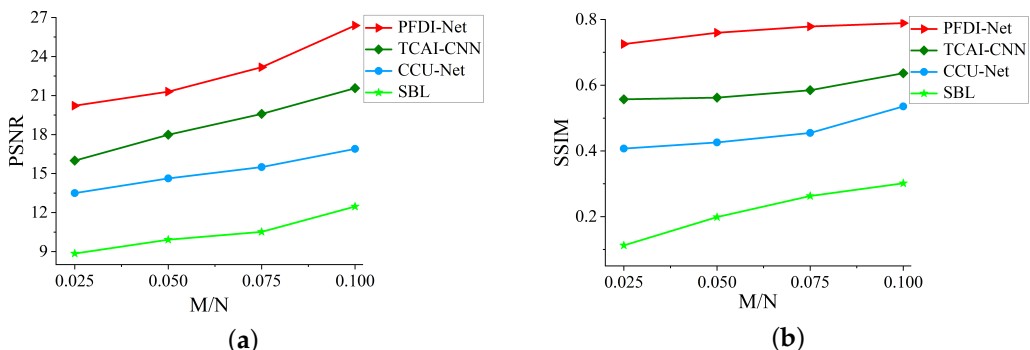

          (**a**)                          (**b**)

**Figure 7.** PFDI-MNIST: Imaging results in two imaging algorithm with different scene information sampling ratio. (**a**) PSNR performance. (**b**) SSIM performance.

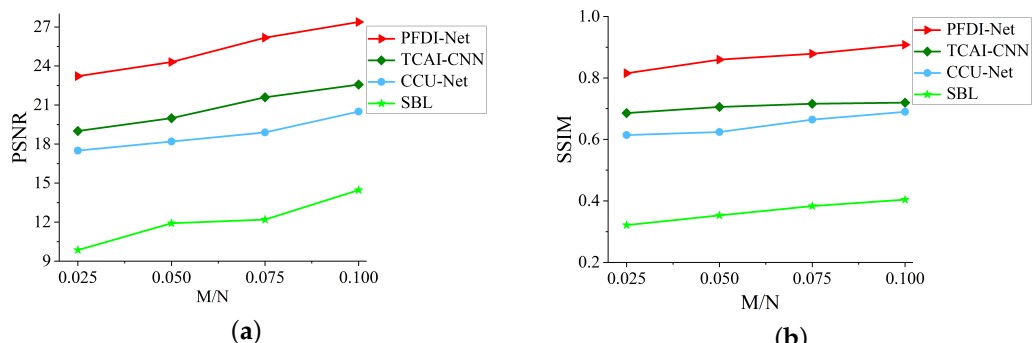

          (**a**)                          (**b**)

**Figure 8.** PFDI-FMNIST: Imaging results in two imaging algorithm with different scene information sampling ratio. (**a**) PSNR performance. (**b**) SSIM performance.

## 4. Discussion

The PFDI-Net algorithm that we proposed displays preferable imaging capacity under noise-free settings compared with the widely used classical MF method for addressing inverse issues and the SBL method using sparse prior information. Imaging mistakes can

be modified adaptively. We incorporate additive noise into the dataset generation process to further assess the proposed algorithm's resilience and noise resistance.

The PFDI-MNIST and PFDI-FMNIST target datasets have scene information sampling rates set to 0.1. We separated the dataset into three noise-free scenarios, namely SNR = 0 dB, SNR = 5 dB, and SNR = 10 dB, after 1000 rounds of testing in order to analyze noise and its effects on network performance. Both datasets are subjected to both qualitative and quantitative assessments. The results of the image reconstruction are displayed in Figure 9. According to the Figure 8, the proposed approach can rebuild not only standard sparse targets but also extended targets under varying SNR and fixed scene compression ratio circumstances. Additionally, when the scene information sampling rate is 0.1, the proposed approach also produces high-quality reconstruction images with improved anti-noise performance and resilience. PSNR and SSIM are quantitatively determined using the proposed approach at various SNR, and the results are displayed in Figure 10.

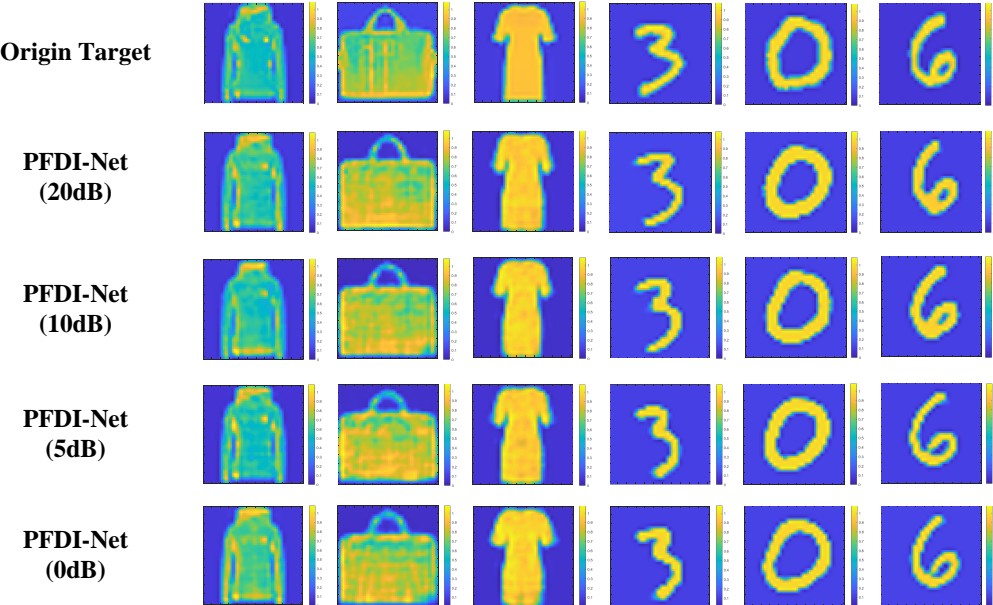

**Figure 9.** Reconstruction results from PFDI-Net with different SNR.

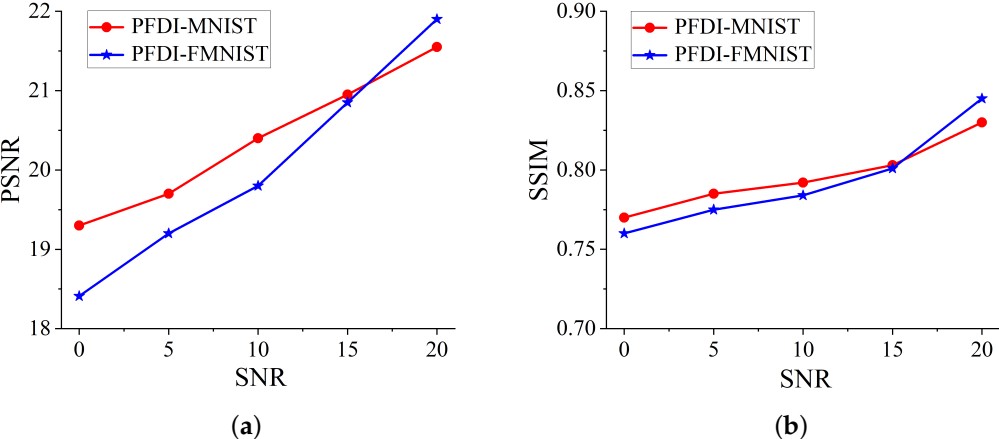

**Figure 10.** Imaging results with different SNR. (**a**) PSNR performance. (**b**) SSIM performance.

These findings demonstrate that our network can be tested with or without noise in the training data with little to no influence on target reconstruction accuracy. In the event of inadequate antenna front-end hardware design, our suggested approach generally compensates for the drawbacks of real-time frequency-diverse imaging. This study highlights

the significant potential of generators using target prior knowledge in the area of phaseless frequency-diverse imaging with good efficiency and resilience.

## 5. Conclusions

This study propose a noniterative method to perform a real-time phaseless frequency-diverse imaging method which incorporates the adversarial generative model with the prior inference model. The intensity-based echo signal can be well resolved through the proposed deep prior generative neural network. Both the inference model and conditional prior model are embedded in the generative model to preferably define the original data space. Simulations results show that the proposed deep reconstruction network can perform near real-time, high-quality scene object reconstruction in both the classical sparse and extended target scenarios, even under extremely low scene sampling ratios and SNR levels, yielding a relatively narrow needed operation frequency band and alleviating the optimal designing burden of current metasurface antennas' front-ends. Moreover, the proposed method has some advantages in terms of imaging efficiency and robustness. With these advantages, this method has potential applications in nondestructive testing, anti-terrorism inspection, and terminal guidance. Future works will focus on the practical imaging demonstrations with the dedicated trained deep reconstruction network and experimentally verifying our method with the measured scene target datasets and echo signals.

**Author Contributions:** Conceptualization, Z.W.; methodology, F.Z.; software, F.Z.; validation, F.Z. and M.Z.; formal analysis, J.Q.; investigation, Z.W. and F.Z.; resources, L.Y.; data curation, M.Z.; writing—original draft preparation, Z.W.; writing—review and editing, F.Z., M.Z., L.Y. and J.Q.; visualization, Z.W.; supervision, L.Y.; project administration, M.Z.; funding acquisition, L.Y. All authors have read and agreed to the published version of the manuscript.

**Funding:** This research was funded by National Natural Science Foundation of China: Grant No. 62201007, U21A20457, 62071003 and China Postdoctoral Science Foundation: No. 2020M681992, and Foundation of An'Hui Educational Committee: No.KJ2020A0026, and Anhui Province University Collaborative Innovation Project: GXXT-2021-028.

**Institutional Review Board Statement:** Not applicable.

**Informed Consent Statement:** Not applicable.

**Data Availability Statement:** The data presented in this study are available on request from the corresponding author.

**Conflicts of Interest:** The authors declare no conflict of interest.

## Abbreviations

The following abbreviations are used in this manuscript:

| | |
|---|---|
| FDI | Frequency-diverse imaging |
| PFDI | Phaseless frequency-diverse imaging |
| CS | Compressed sensing |
| SBL | Sparse Bayesian Learning |
| VAE | Variational Autoencoders |
| GAN | Generative Adversarial Networks |
| PSNR | Peak Signal-to-Noise Ratio |
| SSIM | Structure Similarity Index Measure |
| MSE | Mean Squared Error |
| SNR | Signal-to-Noise Ratio |
| PFDI-MNIST | Phaseless frequency-diverse imaging MNIST |
| PFDI-FMNIST | Phaseless frequency-diverse imaging fashion MNIST |
| PFDI-Net | Phaseless frequency-diverse imaging network |
| ADAM | Adaptive Momentum Estimation |

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
