# Peer review of "Real-Time Phaseless Microwave Frequency-Diverse Imaging with Deep Prior Generative Neural Network"

_remotesensing, doi:10.3390/rs14225665_

Round 1

Reviewer 1 Report

The authors propose a reconstruction network to achieve imaging

in the presence of significant phase errors. They shows few reconstruction results.

However, comparisons have not been shown in the manuscript. Also, they did not review more recent proposals. Therefore, the authors should include more recent methods in the revised manuscript.

Most importantly, the reviewer recommends to compare the proposal with some recent methods and 

explain the advantages of the proposed method over the recent ones.

Moreover, the reviewer suggests to correct grammatically errors.

For example, "Frequency-diverse imaging have gained popularity in both metasurface antennas design and synthetic aperture radar imaging [1–3] applications in recent trends."

Reviewer 2 Report

The paper presents phaseless imaging to relax the phase coherency requirements of imaging system, deep prior generative neural networks are designed to perform scene image reconstruction and the capability of inference network could extract the prior information from the collected echo data and assist the generative nerwork to resolve scene imag. The paper is interesting, however some matters need to be addressed before publication.

1. A better explanation of your contribution needs to be added in the Introduction Section. Please set differences of your work with respect to previous work. Please highligh the advantages, benefits and contributions of the new tecnique and results.

2. Provide more details of the methodology used.

3. There is a lot of paper dealing with this topic. Please cite more references on this topic and considering the applications of the methodology employed.

4. It is needed a better description of the results. Please highligh the benefits, the advantages and contributions of your proposal.

Reviewer 3 Report

In this study authors propose a non-iterative solution for real-time phaseless frequency-diverse imaging in which the GAN ( Generative Adversarial Networks) model with the prior inference model are included. The results show that the intensity-based echo signal can be sucessfully resolved using proposed deep prior generative neural network, in which the inference model and conditional prior model are embedded in the GAN model to preferably define the original data space (for the classical sparse and extended targets scenarios, for extremely low scene sampling ratio and SNR levels).

The several issues should be resolved:

- the contribution should be more clearly defined in introduction section. The conlcusion is very brief - some claims are just statements without clear connection/support in given numerical results without comparison to state-of-the-art methods and/or more detailed discussion (i.e. ... yielding a relatively narrow needed operation frequency band and alleviating the optimal designing  burden of current metasurface antenna front-end).

- the state-of-the-art review is very short and brief. The authors should explain similarities and changes/improvements of their proposal to prior solutions in more details (i.e. to discriminate thier proposal to the current state-of'the-art)

- there is a comprehensive theoretic description of underlying generative models for inference netowrks (the variable markings should be checked to be consistent), but the description of the proposed PFDI-Net solution is a bit unclear. The way the structure is used in the training and testing stage should be separately described. Also, there are references to eq. (10) in text prior to eq. (10) that are a bit confusion (is there a mistake in this referencing). Some markings in text, in fugures and in equations are different (i.e usage of asteriks and ^ for the same value). The description should be more precise and descriptive.

- the proposed approach is compared only with SBLand nnot with ereferent state-of-the-art methods. Since the claim is that this is a new apprach, results support only its applicability, but the comparison to the current state-of-the-art methods is not included, which should be done.

- presentation: Figure 2 is a bit confusing - the structure of neural networks used and other stuctures should be described in text in more details and figure should reflect that. All abbreviations should be given in the text - which is not the case.

Reviewer 4 Report

This paper provides an interesting an end-to-end deep prior generative neural network are designed to achieve near real-time imaging. The topic of this paper is interesting, and the paper is well organized. I think that this paper is suitable for publicating in RS. Stil, I have some suggestions for the authors.

1. The authors use simulated data for GAN traning.  The distributuion of simulated data may deviate from real data distribution if without accurate simutation. Thus, I suggest the authors to mention this point in the paper.

2. The contribution and innovation of the manuscript can be further clearly clarified in abstract and introduction.

3. Please broaden and update the literature review on real applications of deep learning methods in inverse imaging literature.

4. More detailed information about the network layer can be provided, such as such as the number of network layers in each model, the number of filters used in the convolutional layer, and the activation function used by each network layer, etc.

5. There are a few typos and grammatical mistakes in the manuscript, e.g. "Considering the presence of minor systematic errors and alignment errors could still produce heavily corrupted  images." in the abstract.  The verb seems do not agree with its subject.

Round 2

Reviewer 3 Report

The revised version and authors replies to reviewers' concerns are satisfying. The authors introduced adequate changes in the revised version.